# The Neutrophil-to-Lymphocyte Ratio and Preoperative Pulmonary Function Test Results as Predictors of In-Hospital Postoperative Complications after Hip Fracture Surgery in Older Adults [note 1]

**DOI:** 10.3390/jcm12010108

**Published:** 2022-12-23

**Authors:** Seung-Wan Hong, Hae-Chang Jeong, Seong-Hyop Kim

**Affiliations:** 1Department of Anesthesiology and Pain Medicine, Konkuk University Medical Center, Konkuk University School of Medicine, Seoul 05030, Republic of Korea; 2Department of Medicine, Institute of Biomedical Science and Technology, Konkuk University School of Medicine, Seoul 05030, Republic of Korea; 3Department of Infection and Immunology, Konkuk University School of Medicine, Seoul 05030, Republic of Korea

**Keywords:** postoperative complications, hip fracture, older patients

## Abstract

Purpose: This study retrospectively evaluated the usefulness of the neutrophil-to-lymphocyte ratio (NLR), prognostic nutritional index (PNI), and pulmonary function test (PFT) results as objective predictors of in-hospital postoperative complications after hip fracture surgery in older adults. Methods: The patients aged >65 years who underwent hip fracture surgery under general anaesthesia were enrolled. In-hospital postoperative complications with preoperative NLR, PNI and PFT results were evaluated. The NLR was calculated as the preoperative neutrophil count/lymphocyte count in peripheral blood. The PNI was calculated as the serum albumin (g/dL) × 10 + total lymphocyte count × 0.005 (/mm^3^). Results: One hundred ninety nine patients were analysed. The most common postoperative complications were respiratory complications. Compared with patients who did not have postoperative complications, patients with postoperative complications had a significantly higher NLR (8.01 ± 4.70 vs. 5.12 ± 4.34, *p* < 0.001), whereas they had a significantly lower PNI (38.33 ± 6.80 vs. 42.67 ± 6.47, *p* < 0.001), preoperative functional vital capacity (FVC; 2.04 ± 0.76 vs. 2.45 ± 0.71 L, *p* < 0.001), and forced expiratory volume at 1 s (FEV1; 1.43 ± 0.53 vs. 1.78 ± 0.58 L, *p* < 0.001). Multiple logistic regression analysis identified NLR (odds ratio [OR], 1.142; 95% confidence interval [CI], 1.060–1.230; *p* < 0.001) and FEV1 (OR, 0.340; 95% CI, 0.191–0.603; *p* < 0.001) as risk factors for postoperative complications after hip fracture surgery. Conclusion: Preoperative NLR and FEV1 are objective predictors of in-hospital postoperative complications after hip fracture surgery in older patients.

## 1. Introduction

The population of older adults and their medical demands are increasing. Although the current medical status of older adults is much better than in the past, morbidity and mortality after hip fracture surgery remain problematic. Morbidity and mortality after hip fracture surgeries in older patients impose a considerable burden on the patients and, their families and communities [1].

Identification of risk factors for postoperative complications after various surgical procedures has been evaluated. The evaluation has been important to prevent or manage postoperative complications. Among them, neutrophil-to-lymphocyte ratio (NLR) and prognostic nutritional index (PNI) have been used as objective factors. However, their usefulness has not been confirmed with respect to hip fracture surgery in older patients. Moreover, there has been no study focused on postoperative complications during hospital stay [2,3].

Prolonged immobilization due to hip fracture leads to decreased ventilation, decreased expectoration, atelectasis, and pneumonia. Moreover, pulmonary function is decreased by aging. It means that older patients with hip fracture are more vulnerable to deteriorated pulmonary function, and the training for adequate ventilation and cough has been important to prevent the decreased pulmonary function [4]. Therefore, pulmonary function test (PFT) results might also be useful in predicting postoperative respiratory complications. However, no study has evaluated whether PFT results can be used to predict postoperative complications after hip fracture surgery in older patients.

The aim of this study evaluated the usefulness of the NLR, PNI, and PFT results as objective predictors of in-hospital postoperative complications after hip fracture surgery in older patients.

## 2. Materials and Methods

The study was conducted in accordance with the Declaration of Helsinki (as revised in 2013). This retrospective study was approved by the Institutional Review Board of Konkuk University Medical Center (no. KUMC 2022-07-072) and registered at www.cris.nih.go.kr (no. KCT0007469) (accessed on 7 November 2022). Because patient data were obtained from chart reviews, the requirement for informed consent was waived. All extracted data were de-identified before analysis.

### 2.1. Study Population

This study enrolled patients aged >65 years who underwent hip fracture surgery under general anaesthesia with retrospective manner from January 2018 to June 2022 at a single tertiary teaching hospital. A single orthopaedic surgeon performed all procedures. Among enrolled patients, the patients were excluded if they met any of the following criteria: (1) another concurrent fracture, (2) pathological fracture due to cancer or other causes, (3) underlying inflammatory disease, (4) underlying cancer; other concurrent surgery and (5) any other procedure not related to the management of postoperative complications after hip fracture surgery during the hospital stay.

### 2.2. NLR

The NLR was calculated as the preoperative neutrophil count/lymphocyte count in peripheral blood [5].

### 2.3. PNI

The PNI was calculated as the serum albumin (g/dL) × 10 + total lymphocyte count × 0.005 (/mm^3^) [6].

### 2.4. Preoperative PFT

PFT was performed at admission; functional vital capacity (FVC), forced expiratory volume at 1 s (FEV1), and FEV1/FVC were determined.

### 2.5. Anaesthesia Technique and Intra- and Postoperative Management

Perioperative management was standardised and conducted in accordance with the institutional protocol. General anaesthesia was induced with propofol and remifentanil, then maintained with sevoflurane and remifentanil. Anaesthetic depth was maintained during general anaesthesia, using state and response entropies around 50. Rocuronium was administered for muscle relaxation, and peripheral neuromuscular transmission monitoring was performed. During anaesthesia, hemodynamic stability was maintained within 20% of the stability at anaesthesia induction. Neuromuscular blockade was reversed with sugammadex. Postoperative pain was controlled using an intravenous patient-controlled analgesia pump. The decision regarding postoperative admission to the post-anaesthesia care unit or intensive care unit was made via discussion between the attending anaesthesiologist and orthopaedic surgeon, in accordance with the institutional protocol. The orthopaedic surgeon was responsible for postoperative care in a general ward or the intensive care unit, whereas the attending anaesthesiologist was responsible for postoperative care in the post-anaesthesia care unit, in accordance with the institutional protocol.

### 2.6. In-Hospital Postoperative Complications

In-hospital postoperative complications were evaluated, including neurological, respiratory, cardiac, gastrointestinal, hepatic, renal, urinary tract infection, and wound complications, as well as deep vein thrombosis.

Neurological complications included delirium and cerebral infarction. When any new neurological findings were observed, the orthopaedic surgeon checked them and consulted a neurologist or psychiatrist for assessment, diagnosis, and management. Delirium was diagnosed using the confusion assessment method. If necessary, computed tomography or magnetic resonance imaging was performed.

Respiratory complications included atelectasis, pneumonia, pleural effusion, and pulmonary thromboembolism. Postoperative chest radiography was performed in all patients. When new abnormal findings were evident on postoperative chest radiography or new pulmonary findings were observed, the orthopaedic surgeon checked them; performed chest computed tomography; and consulted a pulmonologist for assessment, diagnosis, and management.

Cardiac complications included hemodynamic instability and arrhythmias. Postoperative electrocardiography was performed in all patients. When new cardiac findings were observed, the orthopaedic surgeon checked them and consulted a cardiologist for assessment, diagnosis, and management.

When new gastrointestinal or hepatic complications findings were observed, the orthopaedic surgeon checked them and consulted a gastroenterologist or hepatologist for assessment, diagnosis, and management.

Renal complications, including urinary tract infection, were evaluated using the Acute Kidney Injury Network criteria. When new renal findings were observed, the orthopaedic surgeon checked them and consulted a nephrologist for assessment, diagnosis, and management.

The orthopaedic surgeon checked for deep vein thrombosis using routine computed tomography at 5 days after hip fracture surgery. The orthopaedic surgeon also checked for wound-related complications daily.

### 2.7. Statistical Analyses

Statistical analyses were performed using the Statistical Package for the Social Sciences (SPSS) for Windows, ver. 27.0 (SPSS, Chicago, IL, USA). Demographic data and the NLR, PNI, and preoperative PFT results were analysed according to postoperative complication status. Categorical variables were analysed using the chi-squared test or Fisher’s exact test; continuous variables were analysed using independent t-tests. Univariate analyses of each potential risk factor for postoperative complications were performed using logistic regression. Factors with a *p*-value < 0.05 in univariate analyses were entered into multivariate conditional logistic regression, using the backward stepwise regression procedure. Odds ratios (ORs) with 95% confidence intervals (CIs) were calculated for the final model. To obtain cut-off values for predicting postoperative complications with statistically meaningful factors, the area under the curve was determined using receiver operating characteristic curve analysis with Youden’s index.

Data are expressed as the number of patients, mean ± standard deviation, or median (first-third quartiles). A *p*-value <0.05 was considered indicative of statistical significance.

## 3. Results

During the study period, 210 patients aged >65 years underwent hip surgery. Eleven patients were excluded from the study: six had other concurrent fractures, two had pathological fractures due to cancer, and three had other concurrent surgery. Therefore, 199 patients met the criteria for inclusion in the analysis (Figure 1).

Of the 199 patients analysed, 165 had postoperative complications, including respiratory (*n* = 80), renal (including urinary tract infections; *n* = 29), cardiac (*n* = 15), neurological (*n* = 14), wound-related (*n* = 12), and gastrointestinal or hepatic (*n* = 6) complications, as well as deep vein thrombosis (*n* = 7) (Table 1).

Patients with postoperative complications were significantly older and had significantly higher American Society of Anesthesiologists physical status (ASA PS) classes, compared with patients who did not have complications. Other demographic findings did not significantly differ according to postoperative complication status (Table 2). Patients with postoperative complications also had longer hospital stays (37.3 ± 26.2 vs. 15.1 ± 3.4 days, *p* < 0.001).

Patients with postoperative complications had a significantly higher NLR (8.01 ± 4.70 vs. 5.12 ± 4.34, *p* < 0.001); they had a significantly lower PNI (38.33 ± 6.80 vs. 42.67 ± 6.47, *p* < 0.001) (Table 2), preoperative FVC (2.04 ± 0.76 vs. 2.45 ± 0.71 L, *p* < 0.001), and FEV1 (1.43 ± 0.53 vs. 1.78 ± 0.58 L, *p* < 0.001) (Table 3).

Univariate analyses identified age, NLR, PNI, ASA PS, FVC, and FEV1 as risk factors for postoperative complications after hip fracture surgery. Multiple logistic regression analysis confirmed that NLR (OR, 1.142; 95% CI, 1.060–1.230; *p* < 0.001) and FEV1 (OR, 0.340; 95% CI, 0.191–0.603; *p* < 0.001) were risk factors for complications after hip fracture surgery (Table 4).

The areas under the curve for NLR and FEV1 were 0.746 (95% CI, 0.678–0.814; *p* < 0.001) and 0.709 (95% CI, 0.637–0.784; *p* < 0.001), respectively (Figure 2). The optimal threshold values for predicting postoperative complications based on receiver operating characteristic curve analysis were 5.25 (73.7% sensitivity and 65.4% specificity) for NLR and 1.635 L (63.4% sensitivity and 67.3% specificity) for FEV1 (Figure 2).

## 4. Discussion

This study showed that the NLR and the FEV1 findings in preoperative PFT results could reliably predict in-hospital postoperative complications after hip fracture surgery in older patients, whereas the PNI could not.

Zahorec et al. first presented the NLR in 2001 [7]. They found that the inflammatory response led to changes in the populations of neutrophils and lymphocytes. Therefore, the NLR has been used as an indicator of the inflammatory response in various conditions. A hip fracture produces an inflammatory response, the severity of which might be proportional to the degree of tissue damage around the hip; it may also depend on the patient’s immune status. Several studies have demonstrated that the perioperative inflammatory response is associated with postoperative complications, independent of patient and surgical factors [8,9,10]. Moreover, Chen et al. reported that the inflammatory response could predict in-hospital postoperative complications in patients undergoing total hip arthroplasty [11]. Therefore, we suspected that the NLR might also predict postoperative complications in older patients undergoing hip fracture surgery because of its association with the inflammatory response; our findings supported this notion.

Buzby et al. developed the PNI as a nutritional index in patients undergoing gastrointestinal surgeries in 1980 [12]; Onodera et al. revised the PNI in 1984 [6]. The PNI represents both nutritional and immune statuses because albumin is associated with the nutritional status, whereas the lymphocyte count is associated with the immune status. The PNI has been used to estimate surgical risk and select patients for preoperative nutritional support. Poor nutritional status is an indicator of a poor clinical outcome in various conditions, especially cancer [13,14,15]. Several reports have demonstrated the usefulness of the PNI as a predictor of postoperative complications in older patients undergoing hip fracture surgery [16,17]. Although Ren et al. evaluated the 1-year mortality [16] and Feng et al. evaluated the 5-year mortality [17], no study has examined the association between PNI and in-hospital postoperative complications in older patients undergoing hip fracture surgery. The nutritional status might be an important factor during recovery from hip fracture surgery [18,19]. However, changes in the body that occur before and after hip fracture surgery might be more closely associated with the inflammatory response. Therefore, it might be preferable to measure the degree of the inflammatory response after hip fracture using the NLR [5] because the PNI represents long-term nutritional status, rather than an acute inflammatory response. Based on the use of a mini-nutritional assessment as a diagnostic tool, Malafarina et al. reported that poor nutrition was associated with high mortality after hip fracture surgery in older patients [20]. Although nutritional status is associated with postoperative complications in these older patients, the effect of nutrition on the inflammatory response after hip fracture surgery might be indirect. Therefore, although PNI significantly differed according to the postoperative complication status, it was not statistically significant in multivariate logistic regression analysis.

In this study, the most common postoperative complications were respiratory complications, which might have been the result of immobilisation [21,22]. A hip fracture immediately prevents the patient from walking. This leads to deterioration of respiratory function and exacerbates postoperative complications. Therefore, we presumed that preoperative PFT results might be a useful predictor of postoperative complications in older patients undergoing hip fracture surgery; our findings supported this notion. 

While the ASA PS class significantly differed according to postoperative complication status, it was not statistically significant in multivariate logistic regression analysis. Therefore, although it is widely used, the ASA PS class is not useful as a predictor of postoperative complications after hip fracture surgery in older patients. Actually, several studies have shown that the ASA PS class has limited applications in terms of risk prediction [23,24]. The determination of ASA PS class is dependent on subjective judgement of examiner. Cassai et al. reported that inter-rater reliability for ASA PS class was low to moderate [25].

Comorbid cancer, inflammatory disease and so on had the possibility to influence NLR, PNI and PFT, regardless of hip fracture surgery [26,27]. Therefore, inclusion criteria and exclusion criteria in the present study were strictly established to clearly confirm the association between NLR, PNI and PFT, and in-hospital postoperative complications after hip fracture surgery in older patients and rule out any bias.

This study had some limitations. First, although we described how the NLR represented the acute immune response after hip fracture surgery, we did not measure any immune response biomarkers, such as inflammatory cytokines; however, several studies have demonstrated that a change in the NLR is associated with changes in inflammatory cytokine levels [28,29,30]. Especially, neutrophil has a critical role of innate immune response [31,32]. Web like chromatin structured known as neutrophil extracellular traps (NETs) has been recently focused on the role of immune-mediated condition, including lung injury [33,34,35]. On the consideration of stress condition during perioperative period, the evaluation of thyroid function test (TFT) might be option to replace the measurement of immune response marker in the present study. Thyroid dysfunction, even though euthyroid sick syndrome, was associated with worse outcomes, including more transfusion requirement, after hip fracture surgery [36,37,38]. However, evaluation of TFT was not performed in the present study because TFT was not routinely included as preoperative laboratory test according to the institutional protocol. Second, considering inflammatory process is associated with decreased pulmonary function [39], subgroups analysis might be informative to confirm the correlation between the levels of NLR and PFT, and postoperative complications. However, subgroups analyses were not performed in the present study because the enrolled population was limited. Third, although radiologic evaluation for respiratory pathology was performed preoperatively in all patients, there was still the possibility that a pneumonia might have been missed. In addition, a preoperative DVT might have been also missed on preoperative evaluation. Forth, this was a retrospective study conducted in a single tertiary hospital with a small number of patients. Therefore, the problems associated with retrospective data were unavoidable. However, all surgeries were performed by a single orthopaedic surgeon; thus, this study had the characteristics of a prospective analysis and could overcome the limitations of retrospective investigation. Nonetheless, a well-designed prospective trial should be performed to validate the results discovered in the present study. In addition, different general anesthetic techniques and regional anesthesia should be investigated further as to whether the type of anesthetic correlated with the indicators presently evaluated in this study.

Last, although the length of hospital stay significantly differed according to postoperative complication status, it was not analysed by logistic regression because this would have altered the logistic regression model and influenced the results.

In conclusion, the NLR and the FEV1 findings in preoperative PFT results might be used as objective predictors of in-hospital postoperative complications after hip fracture surgery in older patients.

## Figures and Tables

**Figure 1 jcm-12-00108-f001:**
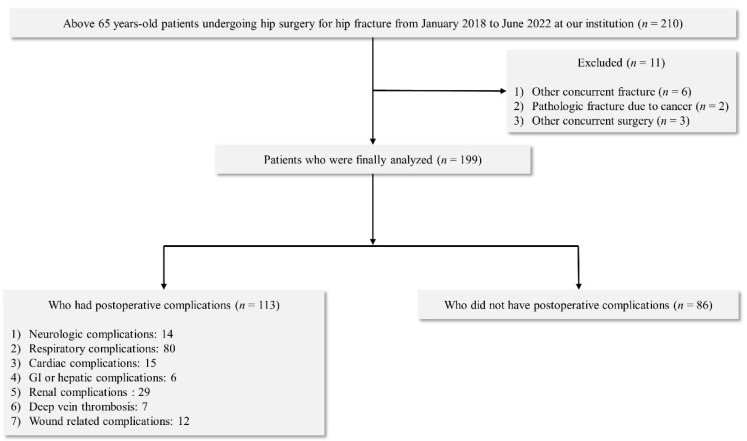
Flowchart of patient selection and classification.

**Figure 2 jcm-12-00108-f002:**
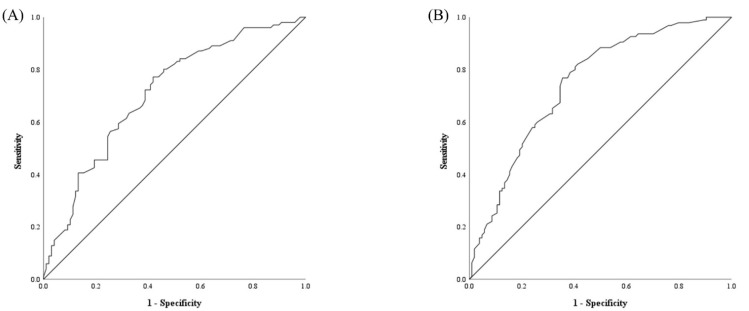
Receiver operating characteristic curves for the neutrophil-to-lymphocyte ratio (**A**) and forced expiratory volume at 1 s during preoperative pulmonary function tests (**B**).

**Table 1 jcm-12-00108-t001:** In-hospital postoperative complications after hip fracture surgery.

Complications		Number of Patients
Neurologic		
	Delirium	13
	Cerebral infarction	1
Respiratory		
	Atelectasis	21
	Pneumonia	17
	Pleural effusion	32
	Pulmonary thromboembolism	10
Cardiac		
	Hemodynamic instability	3
	Arrythmia	12
GI or hepatic		
	Hepatic enzyme elevation	6
Renal		
	Acute kidney injury	12
	Urinary tract infection	17
Deep vein thrombosis		7
Wound related		12

Data is expressed as number of patients. Twenty-three, six, and one patient had two, three, and four types of complications, respectively. Abbreviations: GI, gastrointestinal.

**Table 2 jcm-12-00108-t002:** Demographic data.

		With Cxs (*n* = 113)	Without Cxs (*n* = 86)	*p*-Value
Gender (M/F)		39/74	30/56	0.772
Age (years)		82.3 ± 7.9	79.4 ± 7.2	0.020
Height (cm)		156.1 ± 6.4	156.9 ± 7.3	0.193
Weight (kg)		54.2 ± 11.9	56.2 ± 10.4	0.161
Underlying dz				
	HTN	89	70	0.646
	DM	41	32	0.893
	CVA history	31	27	0.632
	Pulmonary dz	43	29	0.193
	Cardiovascular dz	37	33	0.584
	Renal dz	16	12	0.581
Medication				
	ACEi or ARB	56	34	0.159
	β-blocker	15	7	0.252
	CCB	49	34	0.466
	Diuretics	18	8	0.169
	Hypoglycemic	33	27	0.611
	Antiplatelet	40	27	0.544
ASA PS				0.027
	I	0	0	
	II	54	48	
	III	59	38	
Hospital stay (days)		37.3 ± 26.2	15.1 ± 3.4	<0.001

Data is expressed as number of patients, mean ± standard deviation or median (first-third quartiles). Abbreviations: With Cxs, with postoperative complications after hip fracture surgeries; Without Cxs, without postoperative complications after hip fracture surgeries; M, male; F, female; dz, disease; HTN, hypertension; DM, diabetes mellitus; CVA, cerebrovascular accident; ACEi, angiotensin converting enzyme inhibitor; ARB, angiotensin receptor blocker; CCB, calcium channel blocker; ASA PS, American Society of Anesthesiologists physical status.

**Table 3 jcm-12-00108-t003:** Preoperative laboratory findings and pulmonary function test (PFT) results.

		With Cxs (*n* = 113)	Without Cxs (*n* = 86)	*p*-Value
Laboratory findings				
	NLR	8.01 ± 4.70	5.12 ± 4.34	<0.001
	PNI	38.33 ± 6.8	42.67 ± 6.47	<0.001
PFT				
	FVC (L)	2.04 ± 0.76	2.45 ± 0.71	<0.001
	FEV1 (L)	1.43 ± 0.53	1.78 ± 0.58	<0.001
	FEV1/FVC	70.43 ± 11.76	72.03 ± 8.69	0.290

Data is expressed as number of patients, mean ± standard deviation or median (first-third quartile). Abbreviations: With Cxs, with postoperative complications after hip fracture surgeries; Without Cxs, without postoperative complications after hip fracture surgeries; NLR, neutrophil-to-lymphocyte ratio; PNI, prognostic nutritional index FVC, functional vital capacity; FEV1, forced expiratory volume at 1 s.

**Table 4 jcm-12-00108-t004:** Predictors of in-hospital postoperative complications after hip fracture surgery.

Variables		Univariate		Multivariable	
		Odd Ratio (95% CI)	*p*-Value	Odd Ratio (95% CI)	*p*-Value
Gender (M/F)		1.092 (0.602–1.979)	0.773		
Age (years)		1.046 (1.007–1.088)	0.022		
Height (cm)		0.963 (0.915–1.016)	0.186		
Weight (kg)		1.019 (0.993–1.047)	0.156		
Underlying dz					
	HTN	0.848 (0.418–1.717)	0.646		
	DM	0.961 (0.537–1.779)	0.893		
	CVA history	1.170 (0.615–2.225)	0.632		
	Pulmonary dz	0.642 (0.392–1.134)	0.164		
	Cardiovascular dz	0.649 (0.427–1.184)	0.187		
	Renal disease	0.802 (0.366–1.758)	0.582		
Medication					
	ACEi or ARB	0.665 (0.377–1.173)	0.624		
	β-blocker	0.548 (0.233–1.314)	0.243		
	CCB	0.599 (0.254–1.547)	0.221		
	Diuretic	0.574 (0.232–1.394)	0.433		
	Hypoglycemic agent	1.089 (0.718–2.264)	0.731		
	Antiplatelet agents	0.541 (0.232–1.243)	0.304		
ASA PS		2.113 (1.093–3.957)	0.025		
Laboratory findings					
	NLR	1.159 (1.073–1.252)	<0.001	1.142 (1.060–1.230)	<0.001
	PNI	0.947 (0.906–0.990)	0.001		
PFT					
	FVC (L)	0.476 (0.320–0.707)	<0.001		
	FEV1 (L)	0.328 (0.190–0.566)	<0.001	0.340 (0.191–0.603)	<0.001
	FEV1/FVC	0.985 (0.958–1.013)	0.292		

Abbreviations: dz, disease; HTN, hypertension; DM, diabetes mellitus; CVA, cerebrovascular accident; ACEi, angiotensin converting enzyme inhibitor; ARB, angiotensin receptor blocker; CCB, calcium channel blocker; ASA PS, American Society of Anesthesiologists physical status; NLR, neutrophil-to-lymphocyte ratio; PNI, prognostic nutritional index FVC, functional vital capacity; FEV1, forced expiratory volume at 1 s.

## Data Availability

The datasets are available from the corresponding author on reasonable request.

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
