# Peer review of "The Neutrophil-to-Lymphocyte Ratio and Preoperative Pulmonary Function Test Results as Predictors of In-Hospital Postoperative Complications after Hip Fracture Surgery in Older Adults†"

_jcm, 2022, doi:10.3390/jcm12010108_

Round 1

Reviewer 1 Report

The aim of the study is to evaluate, according to the other studies in the literature, if neutrophil-to-lymphocyte ratio (NLR), prognostic nutritional index (PNI), and pulmonary function test (PFT) results might be usefulness as predictive factors of post-operative complications after hip fracture surgery in adults over 65 yo.

The abstract is comprehensive and exhaustive.

Respiratory post-operative complications are the most frequent and a more exhaustive description about pulmonary function tests (PFT) and about the inflammation process might be interesting.

In this regard, an useful reference is PMID: 32856444 

The methodological approach is correct. Inclusion and exclusion criteria, post-operative complications, all clinical parameters analyzed, were widely explained, and all the data were accurately described.

Therefore, from line 89: among the in-hospital postoperative complications, it might be interesting to argue about the “euthyroid sick syndrome” (ESS).

The results were properly described and summarized in illustrative tables and flow-chart, but the discussion might be more comprehensive.

The conclusions are in line with the purpose of the study.

The study might be worthy of publication after a revision.

A native English speaker might be useful to proofread the text to improve the grammatical content.

Author Response

At first, I thank the editors and referees of the “Journal of Clinical Medicine” for taking their times to review of my paper, entitled “The neutrophil-to-lymphocyte ratio and preoperative pulmonary function test results as predictors of in-hospital postoperative complications after hip fracture surgery in older adults”.

I have made some corrections and clarifications in the manuscript after going over the referee’s comments. The changes are summarized below and the corrected or newly added sentences were expressed with red-color in the manuscript.

Reviewer-1

  1. The aim of the study is to evaluate, according to the other studies in the literature, if neutrophil-to-lymphocyte ratio (NLR), prognostic nutritional index (PNI), and pulmonary function test (PFT) results might be usefulness as predictive factors of post-operative complications after hip fracture surgery in adults over 65 yo. The abstract is comprehensive and exhaustive. Respiratory post-operative complications are the most frequent and a more exhaustive description about pulmonary function tests (PFT) and about the inflammation process might be interesting. In this regard, an useful reference is PMID: 32856444. The methodological approach is correct. Inclusion and exclusion criteria, post-operative complications, all clinical parameters analyzed, were widely explained, and all the data were accurately described. Therefore, from line 89: among the in-hospital postoperative complications, it might be interesting to argue about the “euthyroid sick syndrome” (ESS). The results were properly described and summarized in illustrative tables and flow-chart, but the discussion might be more comprehensive. The conclusions are in line with the purpose of the study. The study might be worthy of publication after a revision. A native English speaker might be useful to proofread the text to improve the grammatical content.

: Thank you for your productive comments. At first, we added some explanations for association between pulmonary function and inflammation in Discussion. Secondly, reviewer recommended to add comment from study conducted by Basilico M et al. They evaluated the risk factors of infection in proximal femur fracture. They checked 122 patients with femur neck fracture and confirmed 15 infections. They found that fever onset within 72 hours from the surgery was statistically correlated with early infections. We thought that the study was conducted with small population and focused on infection. Therefore, the study was not appropriate as a reference for the present study, although reviewer recommended. We are sorry. However, we cited the study conducted by Basilico M et al. for the explanation of the effect of thyroid dysfunction on postoperative complications in hip fracture surgery. Thirdly, thyroid function test (TFT) was not evaluated in the present study because TFT was not included as preoperative laboratory test according to the institutional protocol. However, the change of thyroid function was associated with acute stress condition such as perioperative condition. Moreover, thyroid dysfunction, even though euthyroid sick syndrome, was associated with worse outcomes, including more transfusion requirement, after hip fracture surgery. Therefore, the evaluation of TFT might be informative as reviewer pointed out. We added above in Discussion. Last, the manuscript got English editing service from www.textcheck.com at submission. We added certificate for English editing service.

Reviewer-2

  1. Firstly, the Introduction section does not bring essential information for the Readers who are not extremely familiar with the discussed problem. I strongly suggest expanding it.

: We added essential information for readers to understand easily in Introduction as reviewer recommended.

  1. Did the Authors aim to determine the NLR values after surgery?

: As reviewer knew, the study was retrospectively conducted. Neutrophil-to-lymphocyte ratio (NLR) was calculated, using preoperative laboratory values. We described in Materials and Methods.

  1. What was the rationale to set the exclusions criteria like these? Is there any reference for it?

: We determined inclusion criteria and exclusion criteria to confirm clearly the association between NLR, prognostic nutritional index (PNI) and pulmonary function test (PFT), and in-hospital postoperative complications after hip fracture surgery in older adults. Comorbid cancer, inflammatory disease and so on had the possibility to influence NLR, PNI and PFT, regardless of hip fracture surgery. Therefore, we strictly established inclusion and exclusion criteria to rule out any bias.

 We added above in Discussion.

  1. How long was the postoperative period to be counted into the postoperative complication group?

: As we described in Title and manuscript, postoperative complications were evaluated during hospital stay. The patients with postoperative complications than without postoperative complications, inevitably had the chance have more hospitalization.

We described hospital stay in Table 2 and did not analyze hospital stay, using logistic regression, as described in Discussion.

  1. Was there any approach to identify correlations between basic parameters results and the outcomes?

: Subgroups analysis might be informative to confirm the correlation between the levels of NLR and PFT, and postoperative complications. However, subgroups analyses were not performed in the present study because the enrolled population was limited.

 We added above in Discussion.

  1. Was there any approach to measure other proinflammatory markers and/or markers for neutrophils activation?

: As reviewer pointed out, activation of neutrophils resulted in the changes of inflammation related markers and the measurement of them might be helpful. However, we did not measure it. However, several studies have demonstrated that a change of NLR is associated with changes of inflammatory related markers.

We described above as limitation in Discussion.

  1. Please show in table 2 values with at least one digit after the comma.

: We showed them as reviewer recommended

I hope the revised manuscript will better meet the requirements of the “Journal of Clinical Medicine” for publication.

I thank you again for the constructive review by the referees.

Sincerely yours,

Seong-Hyop Kim, M.D., Ph.D.

Reviewer 2 Report

The presented paper focuses on the potential predictive features of T neutrophil-to-lymphocyte ratio (NLR) and preoperative pulmonary function test (PFT) results as predictors of in-hospital postoperative complications after hip fracture surgery in older adults. 

This topic is pretty interesting since the easy-to-obtain results might have a valuable predictive nature in hospital settings. The area of expertise of the paper also fits well with the scope of the journal. Conclusions are strong and based on the gathered data and results. 

When going through the paper I found these majors that must be fixed: 

Firstly, the Introduction section does not bring essential information for the Readers who are not extremely familiar with the discussed problem. I strongly suggest expanding it. 

Did the Authors aim to determine the NLR values after surgery? 

What was the rationale to set the exclusions criteria like these? Is there any reference for it? 

How long was the postoperative period to be counted into the postoperative complication group?

Was there any approach to identify correlations between basic parameters results and the outcomes? 

Was there any approach to measure other proinflammatory markers and/or markers for neutrophils activation?

Please show in table 2 values with at least one digit after the comma

Author Response

(The authors gave the same response as above.)

Round 2

Reviewer 1 Report

The authors responded satisfactorily to the reviews.

After corrections, the manuscript has increased in quality and is worthy of publication.

Author Response

At first, I thank the editors and referees of the “Journal of Clinical Medicine” for taking their times to review of my paper, entitled “The neutrophil-to-lymphocyte ratio and preoperative pulmonary function test results as predictors of in-hospital postoperative complications after hip fracture surgery in older adults”.

I have made some corrections and clarifications in the manuscript after going over the referee’s comments. The changes are summarized below and the corrected or newly added sentences were expressed with red-color in the manuscript.

Reviewer-1

  1. The The authors responded satisfactorily to the reviews. After corrections, the manuscript has increased in quality and is worthy of publication.

: Thank you for your constructive review.

Reviewer-2

  1. The Authors addressed all my majors and most of the minors in a comprehensive and satisfactory way. The only one more thing that I would like to see is expanding lines 266-268 (discussion about inflammatory response and pulmonary function) by discussing this paper that is extremely relevant: https://doi.org/10.1182/blood.2021014552.

: We added as below with the reference. However, we positioned the paragraph as the first limitation, instead of the second limitation.

→ Especially, neutrophil has a critical role of innate immune response [Nat Rev Immunol 2011; 11: 519-531, Wiley Interdiscip Rev Syst Biol Med 2009; 1: 309-333.]. Web like chromatin structured known as neutrophil extracellular traps (NETs) has been recently focused on the role of immune-mediated condition, including lung injury [Blood 2022; 140: 1020-1037, Front Immunol 2022; 13: 953195, Anesthesiology 2015; 122: 725-727.].

  1. Also, please follow the formatting guidelines of the publisher.

: We re-checked the guideline and corrected the manuscript.

I hope the revised manuscript will better meet the requirements of the “Journal of Clinical Medicine” for publication.

I thank you again for the constructive review by the referees.

Sincerely yours,

Seong-Hyop Kim, M.D., Ph.D.

Reviewer 2 Report

The Authors addressed all my majors and most of the minors in a comprehensive and satisfactory way. 

The only one more thing that I would like to see is expanding lines 266-268 (discussion about inflammatory response and pulmonary function) by discussing this paper that is extremely relevant: 

https://doi.org/10.1182/blood.2021014552

Also, please follow the formatting guidelines of the publisher. 

Author Response

(The authors gave the same response as above.)
